# Tropaeolin OO as a Chemical Sensor for a Trace Amount of Pd(II) Ions Determination

**DOI:** 10.3390/molecules27144511

**Published:** 2022-07-14

**Authors:** Adrianna Pach, Agnieszka Podborska, Edit Csapo, Magdalena Luty-Błocho

**Affiliations:** 1Faculty of Non-Ferrous Metals, AGH University of Science and Technology, A. Mickiewicza Av. 30, 30-059 Krakow, Poland; apach@agh.edu.pl; 2Academic Centre for Materials and Nanotechnology, AGH University of Science and Technology, A. Mickiewicza Av. 30, 30-059 Krakow, Poland; podborsk@agh.edu.pl; 3MTA-SZTE Lendület “Momentum” Noble Metal Nanostructures Research Group, University of Szeged, H-6720 Rerrich B. Sqr. 1, 6720 Szeged, Hungary; juhaszne.csapo.edit@med.u-szeged.hu; 4Interdisciplinary Excellence Center, Department of Physical Chemistry and Materials Science, University of Szeged, H-6720 Rerrich B. Sqr. 1, 6720 Szeged, Hungary

**Keywords:** selective Pd(II) ions determination, tropaeolin OO, metalorganic complex formation, cyclopalladated complex

## Abstract

The selective determination of metals in waste solutions is a very important aspect of the industry and environmental protection. Knowledge of the contents and composition of the waste can contribute to design an efficient process separation and recovery of valuable metals. The problematic issue is primarily the correct determination of metals with similar properties such as palladium and platinum. Thus this paper focuses on the development of a selective method that enables Pd(II) determination in the presence of Pt(IV) ions using the azo-dye tropaeolin OO (TR). For this purpose, the process of the metalorganic complex formation and Pd(II) ions determination were studied by using UV–Vis spectrophotometry under different conditions: solvents (water and B-R buffer), pH (2.09–6.09), temperature (20–60 °C), anions and cations concentrations. The formed metalorganic complex between Pd and tropaeolin OO allows for distinguishing Pd(II) ions from both platinum complexes, i.e. Pt(II), Pt(IV). Moreover, the proposed method can be applied to solutions containing both chloride and chlorate ions. The obtained characteristic spectrum with two maxima allows the determination of palladium even in the presence of other cations (Na, K, Mg, Zn, Co, Ni, Al) and changed concentrations of Pt(IV) ions. Furthermore, the developed spectrophotometric method for the Pd(II) ions determination using tropaeolin OO is characterized by high selectivity towards palladium ions.

## 1. Introduction

In recent years, much attention has been given to platinum-group metals (PGM), due to their unique physical and chemical properties. For these reasons, they are used in many areas of science and industrial processes [1]. Palladium is one of the most recognizable and widely used metals among the PGM. The progressive development of many technological fields caused the increasing use of palladium and its compounds in various areas, including automotive catalysts (55%), electronics (17%), dentistry (9%), and jewellery (11%) [2]. The growing demand for palladium and its compounds may cause release significant amounts of pollution for the environment. Currently, the environmental pollution with palladium in comparison with other toxic heavy metals is minimal. However, growth consumption and, as a consequence, production of more waste can significantly contaminate water and the ecosystem (palladium is not biodegradable), and also may harm human health [3,4]. Therefore, the development of precision methods of recovery and determination of even low amounts of palladium in waste materials and solutions. Currently, the techniques used to determine palladium amounts include methods such as ICP-MS (inductively coupled plasma mass spectroscopy) [5,6,7,8], ICP-AES (inductively coupled plasma atomic emission spectroscopy) [9,10,11], FAAS (atomic absorption spectroscopy) [12,13] and XRF (X-ray fluorescence) [14,15]. Unfortunately, the mentioned methods of determination of palladium have some limitations. These techniques require expensive apparatus, not portable equipment, laboratory facilities, and well-trained personnel. Moreover, the small quantities of Pd and high concentrations of interfering matrix components in samples prevent the direct determination of this metal [16]. The submitted issues cause necessary using various techniques of preconcentration analyzed materials, such as sorption, flotation, and solvent extraction [17].

Currently, more attention is being paid to the spectrophotometric determination of palladium using various organic compounds. The UV–Vis spectrophotometers are commonly available and easy-to-use devices and relatively cheap compared to ICP-MS, ICP-AES, and FAAS techniques. The growing popularity of spectrophotometry is related to the possibility of precise, accurate, and fast analysis materials in comparison to other analysis instruments [18,19]. In the spectrophotometric methods of determination elements, the analyzed compound is carried out in the form of a colored complex, introducing into the system substances that easily form complexes with the determining element (ex. organic substances). The basis for the determination of elements by spectrophotometric methods is the relationship between absorbance and the concentration of the complex formed. The most commonly used complexing reagents are azo compounds. However, they are used for the process of metalorganic complex formation rather than in process of metal determination. The azo-dyes are very interesting organic compounds that have an azobenzene group, which under thermal or photo exposition might lead to *cis-trans* transformation. Compounds such as methyl orange, 2-[-2-(6-nitro benzothiazolyl)azo]- imidazole (NBTAI), HL1 [2-Hydroxy-3-((5-mercapto-1,3,4-thiadiazol-2-yl)diazenyl)-1-naphth aldehyde], HL2 [3-((1,5-Dimethyl-3-oxo-2-phenyl-2,3-dihydro-1Hpyrazol-4-yl)diazenyl)-2-hydroxy-1-naphthaldehyde], 5-(quinoly-8-azo)pyrimidine-2,4,6-trione (L1) and 5-(quinoly-8-azo)-1,3-dimethylpyrimidine-2,4,6-trione (L2), 2-[2−(1-Hydroxy-4-Chloro phenyl) azo]- imidazole (HClPAI) create coloured complexes with many metal ions Co(II), Ni(II), Zn(II), Cd(II), Hg(II), Cr (III), Mn (II), Fe (III), Ru (II) and Pd(II) [20,21,22,23,24,25,26,27].

The main purpose of this study is the development of the spectrophotometric method for the qualitative and quantitative determination of palladium using an azo-dye: tropaeolin OO (TR). Tropaeolin OO (Sodium 4-[(*E*)-(4-anilinophenyl)diazenyl]benzene-1-sulfonate)) is one of the derivatives of methyl orange with an additional phenyl group connected with nitrogen atoms (Figure 1). Tropaeolin OO is not very common in chemistry, but it could be used as pH indicator. In this paper, we present a new application for this compound.

During experiments carried out under different conditions’ colored complexes Pd-TR with a characteristic UV–Vis spectra were formed. Moreover, the efficiency of palladium determination in the presence of metal cations associated with palladium in waste solutions (Na, K, Mg, Zn, Co, Ni, Al) was examined. The Pd determination was possible even in the presence of other precious metal ions: Pt(IV), Pt(II), or popular chloride and chlorate anions.

## 2. Results and Discussion

### 2.1. Experimental Conditions

The process of selective determination of Pd(II) ions was carried out under different conditions (concentration of reagents, pH, temperature, solvents, in the presence of Cl^−^, ClO_4_^−^, Na^+^, K^+^, Al^3+^, Mg^2+^, Co^2+^, Ni^2+^, Zn^2+^, Pt^2+^, Pt^4+^, which have been gathered in Appendix A.

### 2.2. Spectra of Reagents

Before the process of metalorganic complex formation between Pd(II) and tropaeolin OO, all reagents were analyzed using UV–Vis spectrophotometry. Obtained spectra were helpful to fix “a measurement window”, i.e., the wavelength range where the process can be followed without the risk that the spectra coming from substrates, intermediates, and products will overlap.

Registered spectra and graphically determining values of molar coefficient at the characteristic wavelength (λ_1_, λ_2_) for tropaeolin OO were shown in Appendix A and summarized in Table 1.

Depending on the pH of the solution and the type of the solvent (H_2_O, B-R buffer) different spectra were obtained. Generally, for the solution of tropaeolin OO obtained in a buffer solution in the pH range 2.09–6.09 spectra with two strong maxima located at 270 nm and 445 nm, were registered. In the case of lower pH, i.e., 2.09, these maxima were obtained at 207 nm and 460 nm, whereas in aqueous solution these maxima were located at 272 nm and 445 nm (Table 1).

The characteristic spectra and graphically determining values of molar coefficient for Pd(II) in H_2_O solution were shown in Appendix A. The obtained spectra have one strong maximum located at wavelength 209 nm. The value of the molar coefficient equals:λ_209nm_ (ε) = 26547 ± 449 dm^3^·mol^−1^·cm^−1^

### 2.3. Metalorganic Complex Formation between Pd(II) and TR (at Different pH)

The process of complex formation between Pd(II) ions and tropaeolin OO was carried out under different conditions, detailed described in Experimental Conditions (Appendix A. After the reagent mixture in different volume ratios (the same initial reagent concentration) the dynamic change in color was observed.

In an aqueous solution, the orange color coming from tropaeolin OO turn into dark orange, red, pink, and purple within one hour. Finally, after aging (7 days), a green deposit (sample A, Figure 2b), light to dark blue (sample B–D, Figure 2b), dark purple (sample E, Figure 2b), dark red (sample F, Figure 2b) and dark orange (sample G, Figure 2b) were obtained. For samples obtained after 1 h and after aging the UV–Vis spectra were registered and shown in Figure 2a,b. A similar spectra evolution was registered also for different pH (see Appendix A) established using B-R buffer (range of pH 2.09–6.09). The obtained UV–Vis spectra after 7 days and changing samples color from 2 min. to 7 days are shown in Figure 2c–g.

The final color of the samples depends on the applied pH conditions. At the lowest pH = 2.09 samples had red color after reagents mixing (Figure 2c). As the pH increased, the initial color of the solutions changed from orange, pH = 2.87, (Figure 2d) to yellow, pH = 4.1, 5.02, 6.09 (Figure 2e–g). Finally, after 7 days the color of the sample turns light blue, purple, pink, and depending on pH, red (pH 2.09) or yellow (pH > 2.09). Slight color differences can be seen for sample C, which for pH 2.09 is blue, and in higher pH is light purple. Generally, the intensity of the final colors of the samples depends on the pH.

It is worth noting, that at pH = 6.09 the color of the samples obtained at the beginning has stayed almost the same (Figure 2g). In this pH, only samples A–C changed color to light pink after 7 days.

The UV–Vis spectra for the obtained solutions in other pH were registered after reagents mixing, then after 1 h, 24 h which is shown in Appendix A. The final spectra obtained after 7 days are shown in Figure 2c–g. The appearance of obtained spectra is different depending on pH solutions. The UV–Vis spectra obtained after mixing reagents have two characteristics maximum localized at 207 nm, 460 nm for pH 2.09 and 270 nm, 445 nm for another pH, which was related to the original spectrum coming from tropaeolin OO. After 7 days, we noticed the red shift and maxima at 460 nm and 445 nm decreased and new maxima appeared at wavelengths 520–590 nm (depending on the analyzed sample). The most visible changes occur for A–C samples, whereas for D–G samples, the change occurs slowly or not at all (samples G). It also appears the characteristic peak localized at wavelength 703 nm for samples A–C. However, the intensity of these peaks is different, the most visible are for pH 4.10 and 5.02 (Figure 2e,f). The spectrum obtained at the pH = 6.09 changed and also red shift of the maximum was observed (472 nm). The characteristic peak at wavelength 703 nm is not present for pH 6.09 (Figure 2g).

Obtained spectra, registered at a different volumetric proportion of Pd(II) ions to TR and different pH conditions, solvent (see, Appendix A), were used for stoichiometry determination. For this purpose, Job’s plots were created (Appendix A). These plots were determined for the selected wavelength at which the complex has a maximum. There are 540 and 703 nm for pH 2.09 and 2.87 (Appendix A). At higher pH (4.10) Job’s plots were determined at 559 and 703 nm (Appendix A). It is worth noting, that the final peak at pH = 5.02 after 7 days was moved to 718 nm (Appendix A). The constructed Job’s plots suggest that the stoichiometry of the final product (peak localized at 703 nm) is between volumetric ratio Pd(II): TR = 3:1 and 2.5:1.5. The difficulty in unequivocally establishing the molar ratio results from the closeness of the spectra from intermediate and final products. This causes that the individual spectra overlap and do not allow for an unambiguous reading of the values from the Job’s plot. The volumetric ratio 3:1 for the final product and 2:2 for intermediates were suggested in our previous work [20], in which Pd(II) ions form metalorganic complexes with methyl orange at similar experimental conditions.

The obtained results, i.e., analysis of the colors of the samples at different pH (Figure 2), the UV–Vis spectra, suggested that pH = 4.10 and the volumetric ratio 2.5 Pd(II):1.5 TR were suitable for further study due to intensity of the color needed for metal detection. To illustrate how the process of metalorganic changes during the time, evolution spectra at 20 °C were registered and shown in Figure 3a.

It was observed, that after mixing of Pd(II) ions and TR in a volumetric ratio of 2.5:1.5, the change characteristic spectrum for tropaeolin OO with time takes place as was shown in Figure 3a. Decreasing absorbance values at 270 and 445 nm suggests that between Pd(II) and tropaeolin OO reaction proceed. The formation of new spectra with other peaks located at 297, 544, and 703 nm (Figure 3a) suggests that the metalorganic complex appears. This spectrum is similar to that obtained in our previous work [20] and confirms metalorganic complex formation.

To determine the chemical structures of molecules and follow the structural changes in molecules, besides the spectroscopic techniques, the Density Functional Theory (DFT) was used. DFT calculations are extensively used and very helpful to predict the experimental results, especially for organic molecules. DFT calculations were performed to better visualize and explain the changes taking place in the solutions of TR and Pd(II). We assume that in this range of pH the tropaeolin OO is present in anion form and its *trans* conformer is the most stable. However, we should remember that the *cis* form of TR is also present in the solution. According to our previous results for palladium complexes, we predicted the possible structures of TR-Pd complex and optimized their geometry with B3LYP/LanL2DZ method. The optimized structures for tropaeolin OO and its complex with Pd(II) are present in Figure 4.

After structure optimization, the electronic transitions spectra were calculated by using time-dependent TD-DFT with the B3LYP functional and the LanL2DZ basis set. In addition, the electronic spectra were calculated in water solution using the CPCM method. All calculated results were compared with the experimental data and they are in good agreement. (Figure 3b). The calculated spectrum for tropaeolin OO has three peaks: 272, 330, and 480 nm. The last one is connected with the HOMO-LUMO transition (see Appendix A). The theoretical spectrum for structure contains one palladium ion with acetate ligand and is connected with one tropaeolin OO molecule (TR-Pd 1) 324, 347, 485, 546, and 620 nm. This result is very similar to the experimental spectrum obtained after 24 h describe above. In this complex, the HOMO is localized mainly on palladium and acetate ligand, and LUMO is located on tropaeolin OO molecule (see Appendix A). The peak of about 620 nm is connected with HOMO-LUMO transitions. It suggests that after the reaction the metal to ligand charge transfer is observed. After 24 h, the change in the spectrum of the TR-Pd complex was observed. It is caused by the aging of the sample and more complicated structures appear. The third structure TR-Pd 2 presented in Figure 4 is one example of the aging of our complex in which the acetate bridge is forming between two palladium ions connected with the tropaeolin OO molecule. It cannot be said with certainty that our system is not further modified, and the structure consists of more Pd-TR fragments. Experimental spectra measured in a different molar ratio of Pd:TR differ slightly from each other; therefore, it can be concluded with a high probability that other structures are also possible in the tested system. This also explains the slight differences in the experimental and calculated spectrum. The theoretical spectrum was simulated only for a single isolated molecule, whereas the measured spectrum is averaged for all structures present in the solution. Nevertheless, it is clearly visible that as the molecules increase in the structure of the complex, new bands appear. In the case of the TR-Pd 2 structure, two peaks 597 and 728 nm are visible. Both these peaks are related to the HOMO-LUMO transition. HOMO is located mainly on palladium atoms and the acetate bridge, whereas in the case of LUMO, the electron density is transferred to tropaeolin OO molecules (Figure 5).

### 2.4. The Process of Pd(II) Ions Determination

#### 2.4.1. The Influence of Temperature

The important issue in the determination of metal ions is to determine the influence of temperature on to rate of formation of the complex. This paragraph investigated the effect of temperature on the rate of TR-Pd complex formation. The optimal conditions were considered volumetric ratio tropaeolin OO to Pd as 1.5:2.5, the pH value was 4.10. The study was carried out in a UV–Vis spectrophotometer with a thermostatic heated at a different range of temperature 20–60 °C. The UV–Vis spectra obtained after 17.5 h are shown in Figure 6.

The characteristic spectra shift toward higher values for all analyzed temperatures during the process (Appendix A). In the case of all analyzed samples were observed to change solution color from yellow to violet. Regardless of the temperature value, each of the characteristic spectra has four maximum peaks with different absorbance intensities localized in similar wavelengths (Appendix A). The formation of the spectra with the characteristic peak in the wavelength range of 544–573 nm (Appendix A) is responsible for the formation of Pd-N and Pd-C in phenyl ring bond. Finally, the structure TR-Pd 1 (see, Figure 4) is formed. Subsequently, aging of this complex leads to the formation of TR-Pd 2 formation (see, Figure 4) and it is assigned to additional peak position at 703 nm.

As the temperature increases, the maximum intensity of three peaks located at a wavelength: λ_1_, λ_2_, and λ_3_ decreases, simultaneously maximum absorbance value for the peak at λ_3_ increases. As it was expected, the TR-Pd complex formation proceeded faster at increased temperatures.

#### 2.4.2. The Influence of Chloride Ions on the Process of Pd(II) Ions Determination

A very important aspect during the process of Pd(II) ions determination is the presence of chloride ions, which are common ingredients of waste solutions coming from industry. Thus, the process of metalorganic formation between Pd(II) ions and tropaeolin OO was carried out with different addition of chloride ions (see, Appendix A). The obtained spectra and colors of the solutions were shown in Figure 7a,b.

As it was shown, in. Figure 7a, after reagents mixing, i.e., Pd(II) ions and TR in volumetric ratio 2.5:1.5, after one hour only samples contained chloride ions in the range from 0.005 to 0.001 mol/dm^3^ (samples D,E, Figure 7a) reached purple color and spectrum intensity at 703 nm at the same level of absorbance. It suggests that process of metalorganic formation for samples with higher chloride concentration (above 0.01 mol/dm^3^, samples A–C, Figure 7a) takes much more time as was shown in Figure 7b. However, it can be also observed that the primary obtained color of the solution (purple) changes within 24 h into blue (samples D,E, Figure 7b). This, in turn, suggests that the equilibrium between “intermediates” and products changes. It is confirmed by the registered UV–Vis spectrum and its decreasing intensity at 536 nm and increasing intensity at 703 nm. The UV–Vis spectra obtained after 5 min. were shown in Appendix A.

#### 2.4.3. The Influence of Chlorate Ions on the Process of Pd(II) Ions Determination

Taking into account, that waste solutions also may contain chlorate ions, thus the process of metalorganic complex formation between Pd(II) and tropaeolin OO was tested at different amounts of this anion (see, Appendix A). During the process of complex formation, the UV–Vis spectra and colors of solutions were registered. Spectra were obtained after 5 min, 1 h, 24 h since reagents, i.e.: Pd(II) and tropaeolin OO were mixed, as shown in Appendix A.

After one hour, the purple colors of the solutions and similar spectra (except sample F) with characteristic peaks in visible wavelengths located at 538 and 703 nm were registered. Sample F contains a smaller amount of chlorate ions; thus, the process of metalorganic complex formation runs faster. Moreover, the intensity of the peak of this sample at 703 nm is more intensive compared to samples with higher amounts of chlorate anions. Furthermore, the intensity of this peak is the same as that obtained for sample F containing chloride ions, see Figure 7a. It is worth noting, that with time, the colors of samples containing chlorate ions change into blue (except, sample F, see, Appendix A). Finally, after 24 h spectra with intense peaks at 567 nm and 703 nm were obtained. It is worth noting, that the intensity of the peak localized at 703 nm increases, and it is four times more intensive compared to that obtained after one hour (see, Appendix A).

#### 2.4.4. The Influence of the Presence of Other Cations on the Process of Pd(II) Ions Determination

During the process of Pd(II) determination very important aspect is the sensitivity of the proposed method on the metallic impurities. For these reasons, the process of metalorganic formation between Pd(II) ions and tropaeolin OO was carried out in the present of such cations as Zn(II), Co(II), Ni(II), Al(III), Mg(II) and K(I) obtained from dissociation of chlorate salts. After reagents mixing, i.e., Pd(II)/cations with TR the process of metalorganic complex formation takes place. Obtained colors of the solutions and registered spectra were shown in Figure 8 and in Appendix A (Paragraph 8 including spectra registered for all cations after 5 min., 1 h, 24 h at temperatures 20 and 50 °C, Appendix A and colors change with time shown in Appendix A).

The obtained color of the samples and registered spectra suggest that cations such as Na^+^ (see, Appendix A, Sample A), K^+^, Al^3+^, Mg^2+^ (Figure 8a) can be present in the analysed sample, and they do not disturb in process of Pd(II) determination at 536 and 703 nm. In the case of zinc ions, it is also possible. The registered spectrum shown in Figure 8a for Zn(II) ions has characteristic peaks at 542 nm and 703 nm. However, the intensity of these peaks is lower, and it should be taken into account in process of palladium determination. Nickel ions are interesting, because they have green color in the aqueous solutions. These ions have characteristic spectrum with 3 strong peaks located at 394 nm, 543 nm and 718 nm (Figure 8a). After mixing Pd(II)/Ni(II) ions with TR solution, within one hour, the purple color of the solutions was obtained (Figure 8a, Appendix A). The most interesting peak is located at 536 nm and it is responsible for the formation of organometallic compound between Pd(II) and TR. Thus, the process of Pd(II) determination can be also carried out in the presence of Ni ions, but only at one wavelength. The intense pink color of an aqueous solution of cobalt ions also makes the process of qualitative Pd(II) determination problematic. The characteristic maxima of the spectra coming from Co(II) ions and metalorganic complex are in similar region (Appendix A), whereas still the process of quantitative Pd(II) determination is possible through subtracting spectrum coming from metalorganic compound containing Co(II) and spectrum coming from cobalt itself as it was shown in Appendix A).

The process of Pd(II) determination was also carried out in the presence of other PGM cations such as Pt(II) and Pt(IV). Obtained colours of the samples changed with time and spectra evolution were shown in Figure 8b for Pt(II) ions and in Appendix A for Pt(IV). After reagents mixing, i.e., Pd(II)/Pt(II) or Pd(II)/Pt(IV) with TR, the yellow colour coming from azo-dye changes into purple and finally blue with time (Figure 8b, Appendix A). Respectively, the change in spectra were also registered, confirming the process of metalorganic compound between Pd(II) and TR formation. It is worth to note, that Pt(II) and Pt(IV) ions have spectra with maximum at 262 nm. During the process of metalorganic formation only small decrease in intensity of these peaks were observed and it was associated with the process of Pt ions hydrolysis [28].

#### 2.4.5. Applying the Proposed Method in the Practice for Spectrophotometric Determination of Pd(II) Ions

The spectrophotometric determination of Pd(II) ions was realized in the presence of tropaeolin OO and Pt(IV) ions. The process was carried out in a volumetric ratio of reagents was 2.5 mL Pd(II) ions to 1.5 mL TR, pH 4.10 and elevated temperature: 60 °C. Due to the high sensitivity method to initial concentrations of components, the determination was carried out at a wide range of palladium ions and tropaeolin OO concentrations from 5 × 10^−6^ mol/dm^3^ to 2 × 10^−4^ mol/dm^3^. The contents of platinum ions were constant: 5 × 10^−5^ mol/dm^3^.

Before starting the determination process characterized behaviour of the individual reactants: TR and Pd(II) with Pt(IV) in the analyzed concentration range. The intensity of the obtained UV–Vis spectra increases with increasing reagent concentration. The maximum peak for TR has localized in wavelength 445 nm, whereas for a mixture of Pt(II) and Pt(IV) ions the maximum intensity registered at 260 nm (Appendix A).

The UV–Vis spectra after 1h with samples of different colors obtained are shown in Figure 9a. The color of samples varied for different concentrations of reactants. At low concentrations (5 × 10^−6^ mol/dm^3^, 5 × 10^−5^ mol/dm^3^) the solutions were almost colorless, slightly pink for sample B. The solutions with higher concentrations of reagents (1 × 10^−5^ mol/dm^3^, 1 × 10^−4^ mol/dm^3^, 2 × 10^−4^ mol/dm^3^), had a color ranging from light purple-sample C, purple-sample (D) to dark blue for sample F. The obtained color of samples after 5 min, 1 h, and 24 h are shown in Appendix A.

The UV–Vis spectra obtained in the process of Pd(II) ions determination after 5 min and 24 h were shown in Appendix A. However, the registered spectra after 1 h are shown in Figure 9a. The spectra obtained for different concentrations have three maxima located at 260, 560 and 703 nm. The maxima at 560 and 703 nm suggest the formation of metalorganic complex.

The intensity of the spectra increases with higher reagents concentrations and it is consistent with Lambert-Beer’s Law. Based on the spectra (Figure 9a), a graph representing the absorbance versus initial metal ion concentration was constructed and shown in Figure 9b. Designated experimental points provide the opportunity to term molar coefficient for two maximum peaks intensities at wavelength 560 nm and 703 nm. The obtained molar coefficient allows to term exact concentration of the Pd(II) ions in the solution. The absorbance value for the obtained curve fits is represented by two equations:A_560nm_ = 5371 × C_0,Pd(II)_
A_703nm_ = 2866 × C_0,Pd(II)_

The precise determination of the amount of Pd(II) ions in obtained solutions after 1h is possible for the wavelength of 560 nm. Unfortunately, the determination of Pd(II) ions for higher intensity peak 703 nm is difficult, due to low reagents concentrations (5 × 10^−6^ mol/dm^3^, 5 × 10^−5^ mol/dm^3^). Determination of Pd (II) ion at two wavelengths is possible after 24 h, Appendix A. The values of the molar coefficients obtained from the spectral diagrams after 24 h (Appendix A) equal:λ_578nm_ (ε) = 4327 ± 34 mol^−1^·dm^3^·cm^−1^
λ_703nm_ (ε) = 3389 ± 42 mol^−1^·dm^3^·cm^−1^

Generally, can successfully determine the amount of Pd(II) ions using the different concentrations and the appearance of Pt(IV) ions after 1 h. In order to determine of Pd(II) ions at two wavelengths (578 nm and 703 nm), it is suggested to wait for 24 h.

## 3. Materials and Methods

### 3.1. Chemicals

*Tropaeolin (TR)*. The base solution was prepared by dissolving of 0.106 g of TR in 100 mL of deionized water. Next, proper volume of the base solution was diluted in water or in buffers (pH in the range 2.09–7.00), see Table 1.

*Pd(II) chloride complex ions.* The proper volume of base solution of Pd(II) with base concentration 0.093 mol/dm^3^ was diluted in H_2_O or buffers (pH in the range 2.09–7.00), see Table 1.

*Pt(IV) chloride complex ions.* The proper volume of base solution of Pt(IV) with a concentration 0.076 was added to the solution containing of Pd(II) ions.

*Solvents.* In order to keep constant value of pH, the buffer of Britton-Robinson (B-R) was used. For this purpose, 100 mL of acids mixture (0.04 mol/dm^3^ solutions of H_3_PO_4_, H_3_BO_3_ and CH_3_COOH, all reagents were pure analytic, Avantor Performance, Poland) and proper volume of 0.2 mol/dm^3^ solution of NaOH (in the volume range for 7.5 mL for pH 2.09, 17.5 mL for pH 2.87, 25.0 mL for pH 4.10, 35.0 mL for pH 5.02 and 42.5 mL for pH 6.09) was used.

*Anions.* In experiments sodium chloride and chlorate were studied. For this purpose a proper mass of salts were dissolved in B-R solution containing Pd(II) ions (see detail in Table 1).

*Cations.* The source of cations was proper chlorate salts of such metals as Na, K, Co, Ni, Zn, Al, Mg, Pt or chloride salt. All salts were analytical pure and delivered from Avantor Performance, Poland. In experiments 0.1 mol/dm^3^ solution of salts were used. For this purpose, a proper mass of salt was added to the flask (volume of 10 mL) containing Pd(II) in B-R buffer.

### 3.2. Methods of Analysis

Spectrophotometry UV–Vis. The spectra of reagents were registered using spectrophotometer UV–Vis (Shimadzu, Kyoto, Japan), working in the wavelength range 190–900 nm. Device was equipped in reference and thermostatic cell. In all experiments, quartz cuvettes (Hellma, Müllheim, Germany) with optical path 1cm were applied. In standard procedure, a sample was collected immediately after mixture two reacting solutions. Otherwise, the sample was collected and analyzed after 1 h and 7 days. As a reference an aqueous solution of proper solvent was used. All UV–Vis spectra were analyzed using Origin 2021 software.

### 3.3. Density Functional Theory (DFT) Calculation

The structures of palladium complexes were optimized using DFT with the B3LYP hybrid functional, by using a basis set of LanL2DZ, which is very common in palladium complexes calculations. Computations have been performed using the Gaussian 16 program package [29,30,31,32,33].

The electronic transitions were calculated by using time-dependent DFT(TD-DFT) with the B3LYP functional and the LanL2DZ basis set. In addition, the electronic spectra were calculated assuming water as a solvent using the CPCM method [31]. The orbitals and electronic spectra were visualized with GausView5.0. UV–Vis spectra were analyzed using Origin 8.5 software.

## 4. Conclusions

In this work, we show, that process of Pd(II) determination can be carried out at different pH (2.1–6.1), temperature conditions (20–60 °C), in the presence such cations such as Na(I), K(I), Mg(II), Co(II), Ni(II), Zn(II), Al(III) as well as Pt(II, IV) ions and chloride, chlorate anions. For this purpose, the azo compound solution was mixed with palladium ions in order to metalorganic complex formation. It was achieved thanks to the ability of Pd(II) ions to form strong bind to N and C element in phenyl ring of TR, which works as ligand in process of complex synthesis [20]. During these processes, the color changes from orange coming from TR through pink, purple and blue (depending on the experimental conditions and aging time) were observed, giving characteristic UV–Vis spectra being optical response on the complex forms.

It was found that the optimal conditions for carrying out the determination of Pd(II) ions are pH 4.10, temperature 50 or 60 °C and volumetric ratio of Pd(II) ions and tropaeolin OO equals 2.5:1.5. Applied conditions allows for fast response (within one hour) and confirmation of the presence of palladium in the analyte. Moreover, the possible is quantitative (spectrophotometric method, high value of molar coefficient) and qualitative (answer in the form of color) determination of Pd(II) ions in the presence of chloride/chlorate salts containing Zn, Ni, Co, Mg, Al. A very important issue is also the possible determination of the small amount of Pd(II) ions in the appearance of Pt (IV) and Pt (II) ions, which can be realized using the proposed cheap, easily available spectrophotometric method.

## Figures and Tables

**Figure 1 molecules-27-04511-f001:**
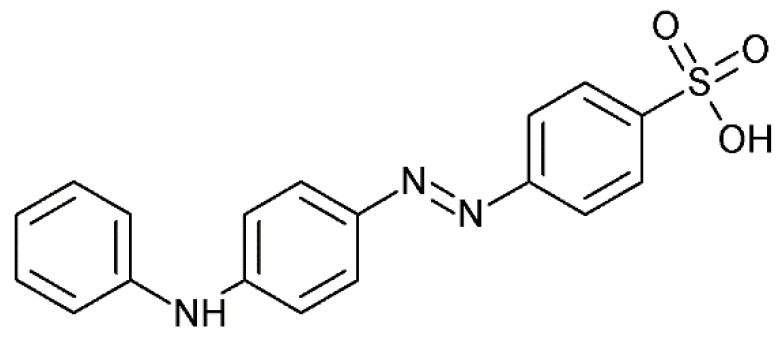
Structure of tropaeoline OO.

**Figure 2 molecules-27-04511-f002:**
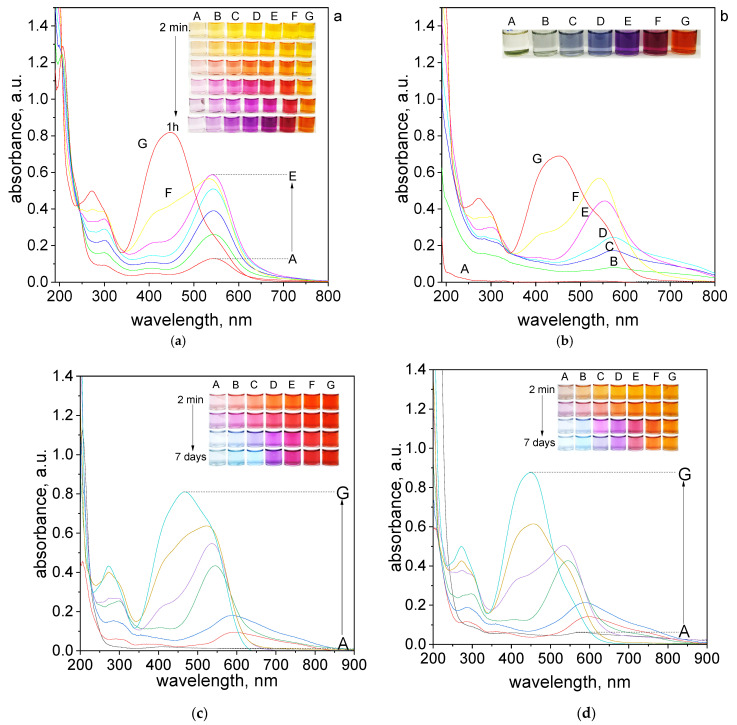
The UV–Vis spectra of solution containing the mixture of tropaeolin OO and Pd(II) at different volume ratio (mL/mL) of Pd(II) to TR (A—0.5:3.5; B—1.0:3.0; C—1.5:2.5; D—2.0:2.0; E—2.5:1.5; F–3.0:1.0; G–3.5:0.5) in H_2_O registered after one hour (**a**) and after 7 days (**b**) and registered after 7 days in B-R buffer, pH: 2.09 (**c**), 2.87 (**d**), 4.10 (**e**), 5.02 (**f**), 6.09 (**g**). Conditions: C_0,TR_ = 5 × 10^−5^ mol/dm^3^, C_0,Pd(II)_ = 5 × 10^−5^ mol/dm^3^ (the value of concentration before reagents mixing), T = 20 °C, path length 1 cm.

**Figure 3 molecules-27-04511-f003:**
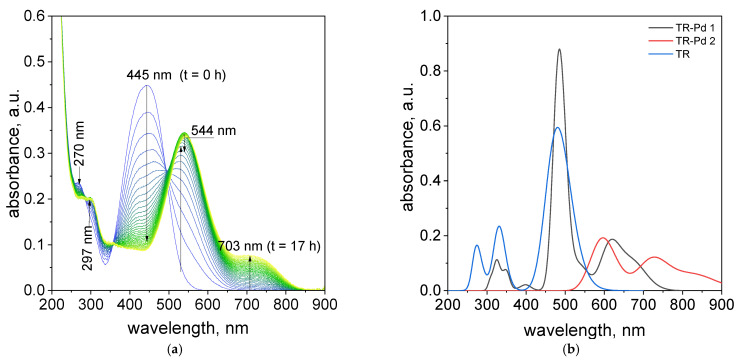
The UV–Vis spectra evolution (24 h) of the solution containing the mixture of tropaeolin OO and Pd(II) at 20 °C. Conditions: volumetric ratio: 2.5 mL Pd(II):1.5 mL TR the value of concentration of Pd(II) ions after mixing with tropaeolin OO: TR = 1.875 × 10^−5^ mol/dm^3^, Pd(II) = 3.125 × 10^−5^ mol/dm^3^, pH = 4.10 (**a**), calculated spectra (**b**).

**Figure 4 molecules-27-04511-f004:**
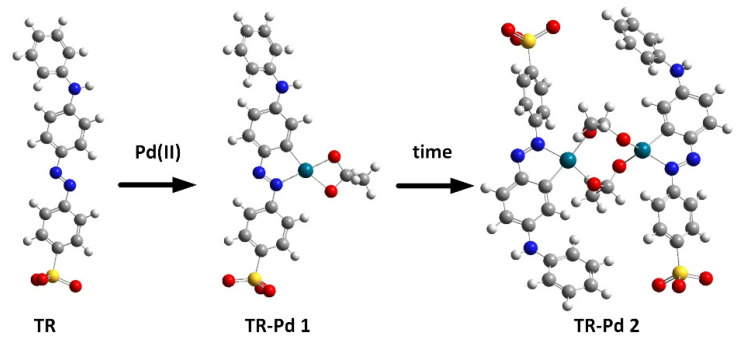
Optimized structure during process of metalorganic complex formation between Pd and tropaeolin OO. Red—oxygen; yellow—sulfur; dark green—palladium; grey—carbon; white—hydrogen, blue—nitrogen.

**Figure 5 molecules-27-04511-f005:**
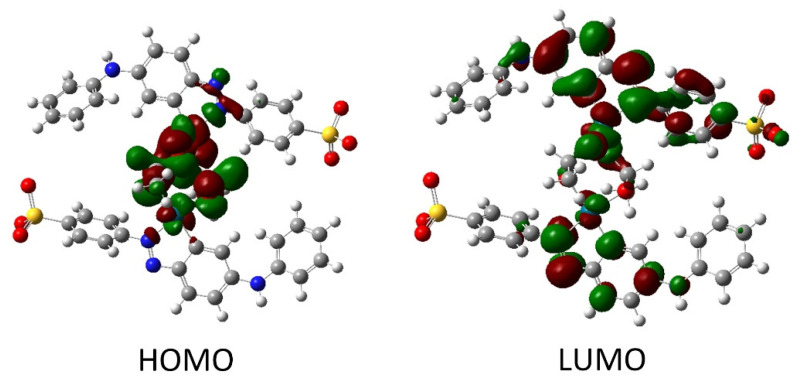
The HOMO and LUMO after DFT calculation for TR-Pd 2 structure.

**Figure 6 molecules-27-04511-f006:**
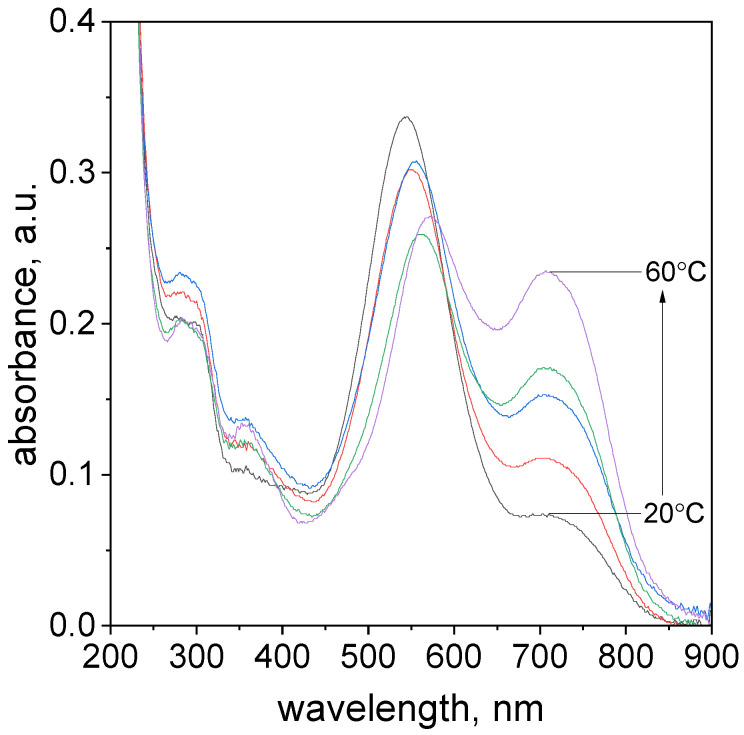
The UV–Vis spectra evolution of the solution containing the mixture of tropaeolin OO and Pd(II) at different temperatures 20–60 °C, absorbance change for all temperatures after 17.5 h. Conditions: volumetric ratio: 2.5 mL Pd(II):1.5 mL TR the value of concentration of Pd(II) ions after mixing with tropaeolin OO:C_0,MO_ = 1.875 × 10^−5^ mol/dm^3^, C_0,Pd(II)_ = 3.125 × 10^−5^ mol/dm^3^, pH = 4.1.

**Figure 7 molecules-27-04511-f007:**
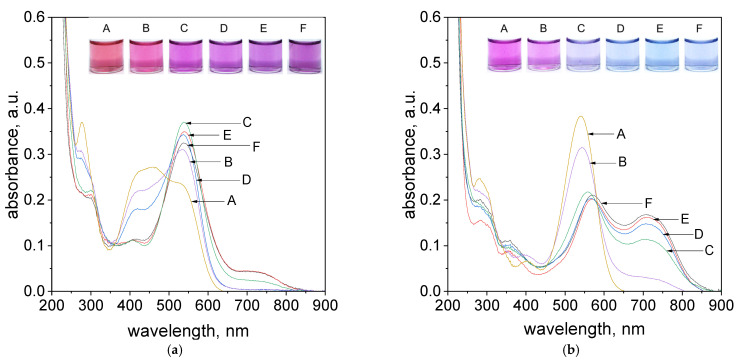
The UV–Vis spectra coming from solution contains the mixture of tropaeolin OO with Pd(II) and different concentrations of Cl^−^, A: 0.1 mol/dm^3^, B: 0.05 mol/dm^3^, C: 0.01 mol/dm^3^, D: 0.005 mol/dm^3^, E: 0.001 mol/dm^3^, F: 0.0005 mol/dm^3^ after 1 h (**a**), 24 h (**b**). Conditions: volumetric ratio: 2.5 mL Pd(II):1.5 mL TR, the value of concentration of Pd(II) ions after mixing with tropaeolin TR = 1.875 × 10^−5^ mol/dm^3^, Pd(II) = 3.125 × 10^−5^ mol/dm^3^, pH = 4.10, temperature 50 °C.

**Figure 8 molecules-27-04511-f008:**
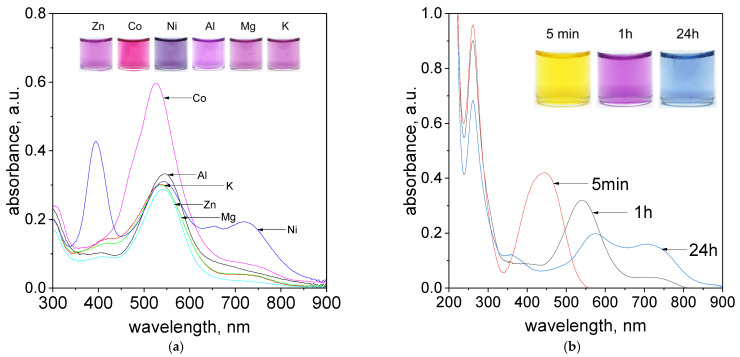
The evolution of UV–Vis spectra solutions containing (**a**) Pd(II) ions, tropaeolin OO and other metal cations: Zn^2+^, Ni^2+^, Mg^2+^, K^+^, Co^2+^, Al^3+^, after 1h at temperature 50 °C, (**b**) Pd(II)/Pt(II) ions and tropaeolin OO, after 5 min, 1 h, 24 h, at temperature 50 °C. Conditions: C_0TR_ = 1.875 × 10^−5^ mol/dm^3^, C_0Pd(II)/Pt(IV)_= 3.125 × 10^−5^ mol/dm^3^ C_0,anions_ = 0.0625 mol/dm^3^, volumetric ratio mixing Pd(II) ions and tropaeolin OO 2.5 mL: 1.5 mL, pH = 4.10.

**Figure 9 molecules-27-04511-f009:**
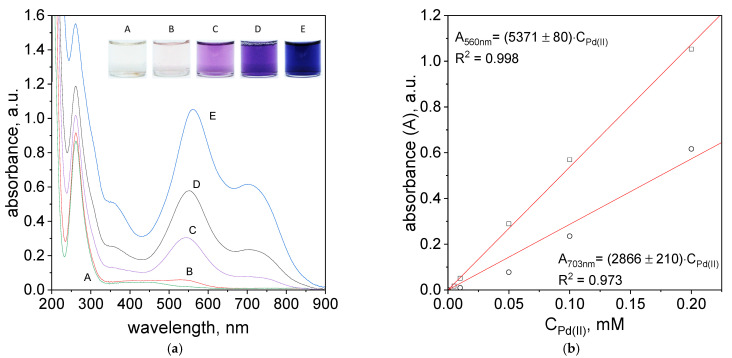
Spectra of solutions containing mixture of TR, Pd(II) and Pt(IV) ions, after 1 h (**a**); dependency of absorbance vs. initial Pd (II) ions, after 1 h, wavelength: 560 nm and 703 nm (**b**). Conditions: C_0TR/C0Pd(II):_ A: 5 × 10^−6^ mol/dm^3^, B: 5 × 10^−5^ mol/dm^3^, C: 1 × 10^−5^ mol/dm^3^, D: 1 × 10^−4^ mol/dm^3^, E: 2 × 10^−4^ mol/dm^3^, C_0Pt(IV)_ = 5 × 10^−5^ mol/dm^3^, volumetric ratio mixing Pd (II) ions and TR 2.5 mL:1.5 mL, pH = 4.10, T = 60 °C.

**Table 1 molecules-27-04511-t001:** The values of wavelength and molar coefficient (ε) at different pH of the tropaeolin OO solution, T = 20 °C.

pH (Solvent)	ε_1_ (λ_1_)	ε_2_ (λ_2_)
	M^−1^·cm^−1^	M^−1^·cm^−1^
4–5 (H_2_O)	11,458 ± 130 (272 nm)	22,350 ± 59 (445 nm)
2.09 (B-R)	6506 ± 54 (207 nm)	12,616 ± 53 (460 nm)
2.87 (B-R)	7491 ± 200 (270 nm)	14,793 ± 129 (445 nm)
4.10 (B-R)	7652 ± 184 (270 nm)	15,272 ± 123 (445 nm)
5.02 (B-R)	7369 ± 181 (270 nm)	15,341 ± 114 (445 nm)
6.09 (B-R)	7981 ± 36 (270 nm)	15,850 ± 72 (445 nm)

## Data Availability

Not applicable.

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
