# Peer review of "Tropaeolin OO as a Chemical Sensor for a Trace Amount of Pd(II) Ions Determination"

_molecules, 2022, doi:10.3390/molecules27144511_

Round 1

Reviewer 1 Report

See attached file 

Author Response

Reviewer 1

The paper is interesting and faces a actual problem of selective detection of metals with similar properties.

However, the language needs a serious revision, before it can be considered for publication. The paper is riddled with grammar mistakes that make it difficult, at times, to understand the text.

Response: The text was corrected.

In general, it should be shortened, better structured and the definitions should be more precisely. For instance, the volume ratio is imprecisely indicated as stoichiometry.

Response: According to the reviewer’s suggestion, Tables 1, 3 were moved to SI. We changed also sentences that suggest that the volumetric ratio corresponds to stoichiometry.

The description of the method should not be in the section of the Results.

Response: Methods were described in a separate paragraph.

Table 1: The units should be homogeneous. Furthermore, it should be more compact.

Response: The units were corrected.

Table 3 can be eliminated, and the main issues summarized in the text.

Response: Table 3 was moved to SI.

The presentation of the kinetics effects are confusing, since at start the temperature is mentioned, hinting at thermodynamics issues. In general, the plot of the peaks maxima as a function of time would give a better overview of the processes.

Response: The Fig. 2 was improved. Now, it shows, that with time, during the process of complex formation between Pd and TR, the change in the intensity of the spectrum at about 450 nm and increasing at 560 and 703 nm, was observed.

The color variation with time is rather slow, and it looks like if the method is not suited for practical applications. Can you envisage a device based on the system you proposed?

Response: The Reviewer is right, that the proposed method is slow (response within 1 hour). However, if we take into account, that in the case of a huge factory, the analytical laboratory is usually in another building thus samples have to be first delivered. Moreover, a huge advantage of the proposed method is the fact that it is selective and it allows for qualitative and quantitative Pd(II) determination, even in the presence of Pt(II, IV), which is a major problem.

Finally, why did you choose the Britton Robinson buffer?

Response: We choose, B-R buffer, due to its composition. It contains acetic acid, which allows for acetic bridge formation during the process of complex dimerization and has an influence on complex stabilization. Experiments carried out in water (pH 4 – 5) shows, that in some cases we get deposit on the bottom of vials (Fig. 1b).

Reviewer 2 Report

Comments to the Author:

In the manuscript titled Tropaeolin OO as a chemical sensor for a trace amount of Pd(II) ions determination, authors focused on the development of a selective method that enabled Pd(II) determination in the presence of Pt(IV) ions using the azo-dye tropaeolin OO (TR) and developed spectroscopic method for the Pd (II) ions determination using tropaeolin OO is characterized by high selectivity towards palladium ions

I have read the manuscript carefully and I have following queries. Therefore, I request to address the following comments:

Comments:

1.      Structure of Tropaeolin OO is not given. It must be added before discussion.

2.      Figure 1, superscript should be written properly.

3.      Lines 364, 366, units should be written properly.

4.      If the jobs plot was taken kindly add it in the main article or add a discussion about it that at which ratio ligand to metal is complex is formed.

5.      Discuss the UV-Vis peaks corresponding to transitions in the structure.

6.      Why 1.5:2.5 ratio has been chosen, this is misleading as per the DFT the complex ratio is 2:1 (ligand: metal)

7.      In the temperature experiment, assign the peaks to the transition that occurred in the complex to show the change in transitions due to temperature (assign the atom of molecules linked with metal ion)

8.       As discussed in the conclusion, no bonding of Pd(II) has been shown with N and C atoms of TR (give structure in discussion)

9.      Practical applicability of the work should be mentioned as it has been standardized at 4.0 pH

10.  This work showed a complete change in color after an hour which is a very long time for sensitive detection.

The manuscript can be accepted for publication after addressing these comments.

Author Response

Reviewer 2

Comments to the Author:

In the manuscript titled Tropaeolin OO as a chemical sensor for a trace amount of Pd(II) ions determination, authors focused on the development of a selective method that enabled Pd(II) determination in the presence of Pt(IV) ions using the azo-dye tropaeolin OO (TR) and developed spectroscopic method for the Pd (II) ions determination using tropaeolin OO is characterized by high selectivity towards palladium ions

I have read the manuscript carefully and I have following queries. Therefore, I request to address the following comments:

Comments:

  1. Structure of Tropaeolin OO is not given. It must be added before discussion.

Response: Now, the structure of tropaeoline OO was shown in Fig. 1.

  1. Figure 1, superscript should be written properly.

Response: It was corrected.

  1. Lines 364, 366, units should be written properly.

Response: It was corrected.

  1. If the jobs plot was taken kindly add it in the main article or add a discussion about it that at which ratio ligand to metal is complex is formed.

Response: Based on Job’s plot it is difficult to clearly define stoichiometry. The difficulty in unequivocally establishing the molar ratio results from the closeness of the spectra from substracts, intermediate and final products. This causes the individual spectra to overlap and do not allow for an unambiguous reading of the values from the Job’s plot. It was commented on in the manuscript.

  1. Discuss the UV-Vis peaks corresponding to transitions in the structure.

Response: This aspect was described on pages 9 and 10.

  1. Why 1.5:2.5 ratio has been chosen, this is misleading as per the DFT the complex ratio is 2:1 (ligand: metal)

Response: We select such an experimental molar ratio because our aim was to use the process of complex formation as a tool for Pd(II) ions determination.  It was observed that at this proportion of ligand to metal the intensity of the color of the formed complex was optimum, which was crucial to the process of metal determination. In the case of DFT calculations, the ratio ligand to metal was 1:1 or 2:2 depending on the stage of our experiment. DFT calculation helped us to determine the bond between ligand and Pd ions. For calculations, we can use one isolated structure but in an experiment condition, we have a mix: structures 1:1, 2:2 (presented in Fig. 4), rest of the unreacted substrates: tropaeolin OO and palladium complexes.  That way these two results are not in very good agreement.

  1. In the temperature experiment, assign the peaks to the transition that occurred in the complex to show the change in transitions due to temperature (assign the atom of molecules linked with metal ion)

Response:  According to the Reviewer's suggestion, in paragraph  2.4.1. we assigned the peaks to the transition that occurred in the complex. The following sentences have been added:

The formation of the spectra with the characteristic peak in the wavelength range of 544 – 573 nm (Supporting Materials, Table S2) is responsible for the formation of Pd – N and Pd – C in the phenyl ring bond. Finally, the structure TR – Pd 1 (see, Fig. 4) is formed. Subsequently, the aging of this complex leads to the formation of TR – Pd 2 formation (see, Fig. 4) and it is assigned to an additional peak position at 703 nm.

  1. As discussed in the conclusion, no bonding of Pd(II) has been shown with N and C atoms of TR (give structure in discussion)

Response: The structure formed during the process of metalorganic complex formation between Pd and tropaeolin OO was shown in Fig. 4. Information about Pd – N and Pd – C in phenyl ring bond formation was added in paragraph 2.4.1.

  1. Practical applicability of the work should be mentioned as it has been standardized at 4.0 pH

Response: The choice of pH = 4.1 was related to the buffer composition, similar results will be obtained at pH = 4.0. The described method is working in a broader pH range, i.e. 2 – 5.

  1. This work showed a complete change in color after an hour which is a very long time for sensitive detection.

Response: On one hand, the Reviewer is right, but on the other no. Exemplary, if we take into account, that we need only response YES/NO, it means quality information about the presence of Pd(II) ions, thus this time is quite long. However, if we take into account quantification, this time (one hour) is enough to carry samples from factory to laboratory in order to proper Pd(II) determination.

The manuscript can be accepted for publication after addressing these comments.

Round 2

Reviewer 1 Report

The authors properly amended the manuscript.